# Parametric Study of the Influence of Nonlinear Elastic Characteristics of Rail Pads on Wheel–Rail Vibrations

**DOI:** 10.3390/ma16041531

**Published:** 2023-02-12

**Authors:** Traian Mazilu, Mădălina Dumitriu, Ionuț-Radu Răcănel

**Affiliations:** 1Department of Railway Vehicles, University Politehnica of Bucharest, 060042 Bucharest, Romania; 2Department of Strength of Materials, Bridges and Tunnels, Technical University of Civil Engineering Bucharest, 020396 Bucharest, Romania

**Keywords:** wheel–rail vibration, rail pad, bilinear function, Euler-Bernoulli beam, inhomogeneous foundation

## Abstract

The rail pad is the elastic element between the rail and the sleeper that has the role of absorbing the mechanical stresses from the rail and reducing the vibrations and shocks generated by wheel–rail interactions. In this paper, the problem of the influence of the variability of the nonlinear load-deformation characteristic of rail pads (resulting from the manufacturing process) on wheel–rail vibrations is investigated. The limit load-deformation characteristics of a manufactured rail pad and the medium load-deformation characteristic resulting as the arithmetic mean of the two are considered. The nonlinear load-deformation characteristic of the ballast is also considered. All these characteristics are approximated with the help of the bilinear function and are implemented in a track model consisting of an infinite Euler-Bernoulli beam placed on a two-elastic layer continuous foundation with inertial insertion, resulting in a model with an inhomogeneous foundation. The parameters of the inhomogeneous foundation are established from the equilibrium condition under a static load. Wheel–rail vibrations are studied in terms of the contact force and the acceleration of the rail and wheel. The influence of the variability of the elastic characteristics of the rail pad manifests itself in the field of medium frequencies, which amplify or attenuate the vibration levels in certain bands of one-third of an octave.

## 1. Introduction

The rail pad is a piece of elastic material comprising part of the fastening system that includes other components that provide a structural link between rail and sleeper (Figure 1 based on [1]). Depending on the type of fastening system, namely, a direct (Figure 1a) or indirect system (Figure 1b), the rail pad is inserted between the rail foot and the sleeper or includes a steel baseplate between the rail pad and sleeper which is fixed to the sleeper using other fasteners than those used to fix the rail by the steel baseplate.

The main roles of the rail pad are to elastically absorb the force from the rail by transmitting it to the sleeper and to damp the vibrations and shocks caused by traffic. To this end, rail pads are typically fabricated from resilient materials like rubber, ethylene propylene diene monomer (EPDM), thermoplastic polyester elastomer (TPE), ethylene-vinyl acetate (EVA) or high-density polyethylene (HDPE) [2,3]. Some of these rail pads are presented in Figure 2. 

The load-deformation of the rail pad has a specific shape dictated by the way it must behave on the track under the action of the force coming from the rail. The rail pad stiffness must be small at the beginning so that the deformation under the action of the clamping force is large enough to ensure intimate contact with the rail foot, regardless of the vertical movements of the rail. When a train wheel is above the rail pad, its stiffness must be high to prevent large movements that could lead to a weakening of the grip. Such an elastic characteristic can be achieved by choosing an appropriate fabrication material and, if necessary, by profiling one side of the pad (see, Figure 2a,c). 

It follows from the above that rail pads have nonlinear, quasi-static elastic characteristics, as described in several works [3,5,6]. This nonlinear characteristic of the rail pad, together with the non-linear elastic characteristic of the ballast and subgrade [5,7,8], determine the specific shape of the load-deformation curve of the track [9,10,11,12]. Additionally, the dynamic behaviour of the rail pad exhibits nonlinear aspects in terms of the frequency response function [3,13,14,15,16,17,18]. 

As a component of the track, the fastening system containing the rail pad is a critical element which needs to be included in models when studying problems of practical interest, such as those related to rolling noise, the mitigation of the vibrations in soil, wear of rolling surfaces, etc.

Mechanical representations of tracks are based on the continuous medium theory, whose equations can be resolved by analytical, semi-analytical or numerical methods. In the case of analytical and semi-analytical methods, the rail can be modelled using Euler-Bernoulli beam theory [19,20] or Timoshenko beam theory [21,22]. There are two approaches: either the beam is finite, in which case the modal analysis method is applied to obtain time-domain simulations [23], or the beam is infinite, which has the advantage of eliminating the effect of the waves which are reflected from the edges of the model [24]. 

The main numerical methods applied for track modelling are the finite element method, the boundary element method and the discrete element method. The finite element method is recommended for track modelling in the domain of high frequencies, because this method offers more possibilities in terms of reproducing constructive details. When the finite element method is used, the length of the model is always finite, and for this reason, the method exhibits the disadvantages associated with the finite length model mentioned above. The calculation time is considerably higher because the models have a very large number of finite elements and nodes [25,26]. A direction of research in which the finite element method is applied is that of rolling noise. In this case, the finite element method is supplemented with the boundary element method to calculate the sound pressure level [27,28]. 

The discrete element method subsumes any numerical method that allows the calculation of the motion of a large number of small bodies that are in contact or collide with each other or with a surface. The method has many applications, especially in problems involving granular materials such as the ballast. The use of this method is recommended to solve the problems concerning the specific phenomena of the ballast that lead to track unevenness, i.e., the migration of the ballast, resulting in hanging sleepers, and ballast settlement [29,30].

Regarding rail pad modelling, the simplest representations are linear models with concentrated parameters of the Kelvin-Voigt [25,31] or Poynting-Thompson [14] types, or with distributed parameters, i.e., of the Winkler type with viscous damping [23]. In [32,33], the rail and the sleepers are modelled using the finite element method, and the rail pad is modelled with the help of several discrete Kelvin-Voight systems that are regularly distributed on a line segment or on a rectangular surface. Obviously, these models do not reflect the nonlinearity of the load-deformation characteristic of the rail pad. Further, the loss factor is proportional to the frequency, which limits the frequency up to which they can be used. Better results from this point of view are obtained when hysteretic damping is introduced instead of viscous damping [34]. 

Other techniques with which to study the viscoelastic features of rail pads include the use of the fractional derivative model [35].

The nonlinear load-deformation characteristics of rail pads can be modelled using polynomial functions [36] or the bilinear function [37,38].

In this paper, the impact of the load-deformation characteristic of the rail pad on the wheel–rail vibration behaviour is treated. Specifically, we aim to highlight the impact of the variability of this characteristic, resulting from the manufacturing of the pads. For instance, Figure 3 shows the load-deformation characteristic for a certain rubber rail pad used at CFR (Romanian Railway), as well as the two limits, namely, the stiff limit and the soft limit, within which this characteristic must fall [39]; otherwise, the rail pad is rejected. It is observed that there is a wide margin between the two limit characteristics, and the question is asked: What is the potential impact on the wheel–rail vibration behaviour and, in turn, on the functionality of the rail pad? Obviously, narrowing the accepted tolerance margin leads to an increase in the number of rejected rail pads and to an increase in manufacturing costs, while widening this margin can have negative effects on the functionality of the rail pads.

To deal with this problem, a wheel–rail interaction model was developed in which the wheel is represented by a rigid body with one degree of freedom loaded by the vehicle weight on a wheel. This simple wheel model was chosen to limit the number of parameters in order to highlight more clearly the impact of the load-deformation characteristic of the rail pad. Such an approach is frequently encountered [20,22,23,34,37]. As for the track, our model considers the nonlinear load-deformation characteristics of the rail pad and ballast, as represented by bilinear functions [37,38]. The rail equilibrium position under a static load on the wheel was first determined and the elastic load-deformation characteristics of the rail were obtained. The wheel–rail interaction model is that of roughness displacement [22], and the rail receptance was calculated using a model with a nonhomogeneous foundation, whose parameters result from the equilibrium position. The impact of the variability of the rail pad load-deformation characteristic on wheel–rail vibrations in terms of dynamic contact force, wheel speed, and rail speed at the contact point was evaluated.

To the knowledge of the authors, the influence of the variability of the elastic characteristic of the rail pad on wheel–rail vibrations has not been treated in the past in the manner described in this paper.

## 2. Mechanical Model and Governing Equations

### 2.1. Hypotheses and Structure of the Mechanical Model

The modelling of a wheel–rail system to investigate the behaviour of the vertical vibrations is based upon several hypotheses that are widely encountered in the specialized literature. Thus, the vehicle and the running track are considered symmetrical in relation to the median longitudinal plane. Moreover, to operate railway vehicles, they must have balanced wheel loads, both when empty and when loaded. As a result, the two rails of the track are equally stressed during running; a similar observation can be made regarding the wheels of the vehicle. Under these conditions, it is expected that the wear of the running surfaces takes a reasonably similar shape on the two sides of the vehicle–track system. Consequently, the vehicle–track system can be reduced to half, taking the longitudinal vertical plane as the separation plane. In other words, vertical vibration studies consider a 2D mechanical model that represents half of the vehicle and one rail. 

The following assumption refers to the frequency domain, in which the natural frequencies of both the vehicle and the wheel–rail system are located. The vehicle’s natural frequencies are between 0.5–20 Hz, and the corresponding vehicle vibration modes are excited by irregularities in the track superstructure with wavelengths between 3 and 100–120 m. The wheel–rail natural frequencies are greater than 30–40 Hz, and the vibrations, which can reach 2–3000 Hz, are induced by the irregularities of the rolling surfaces, with wavelengths between 2–4 cm and 3 m. From the two facts presented above, it follows that the vehicle vibration and wheel train vibration are decoupled, and that the wheels vibrate independently of each other, because the coupling resulting from the suspension is unimportant. The vehicle model is thus reduced to a train of loaded wheels running on a rail; the load is actually the static load corresponding to a wheel.

The third assumption is that the rail bending waves generated by the vibration of a wheel–rail pair will have a negligeable effect upon the vibration of other wheel–rail pair. Indeed, depending on the frequency, the rail bending waves have two or one evanescent waves and zero or one propagative wave. Evanescent waves are strongly attenuated by their very nature, and the attenuation of the propagative waves is reduced as the frequency increases. Since the present study deals with low and medium frequencies, it follows from the above that the coupling between the vehicle wheels due to rail bending waves can be neglected.

Adopting the above assumptions, the mechanical model shown in Figure 4 is proposed to analyze the wheel–rail vibrations. The vehicle model is reduced to a wheel which is considered as a rigid body of mass (*M_w_*) loaded with the static load *Q*, and that of the track is simplified to a rail represented by an infinitely uniform Euler-Bernoulli beam on an elastic two-layer foundation between which an inertial layer is interposed [40]. The elastic layers model the effect of the rail pads and the ballast, while the inertial layer models the influence of the sleepers on the rail vibrations. The frequency domain of this type of model is limited to 6–700 Hz, because the effect of the spacing of the sleepers on the rail bending vibrations (the pinned-pinned mode) is neglected. The track model parameters are the bending stiffness, *EI*, and the mass per length unit *m* of the rail, the stiffness per length unit *k*_1*e*_ and *k*_1*r*_ for the rail pad, the mass per length unit *m_s_* of the sleepers, and the stiffness per length unit *k*_2*e*_ and *k*_2*r*_ for the ballast. Stiffness *k*_1*e*_, *k*_1*s*_, *k*_2*e*_ and *k*_2*s*_ are determined by bilinear approximations of the nonlinear load-displacement characteristics of the rail pad and ballast. Distances *l*_1_ and *l*_2_ are determined from the equilibrium condition under static load *Q*.

### 2.2. Bilinear Approximation of the Load-Displacement Characteristics of the Rail Pad and Ballast

Figure 5 displays the stiff and soft limits of the load-displacement characteristics of the rail pad and the mean characteristic calculated as the arithmetic mean of the limit characteristics. Typically, these three characteristics are associated with stiff, medium, and soft rail pads. Additionally, the nonlinear load-displacement characteristic of the ballast per semi-sleeper is presented. This characteristic obeys Hertz’s law [5]
(1)$$Q=C_{H}\sqrt{u_{b}^{3}},$$
where *Q* is the load, *u_b_* is ballast deformation, and *C_H_* is the Hertz constant.

Each nonlinear characteristic is approximated by a bilinear function which has elastic and rigid parts (Figure 6)
(2)$$Q(u_{r})=\left\{\begin{array}{cc} k_{re}u_{r} & 0<u_{r}<u_{ro}\\ k_{re}u_{ro}+k_{rr}(u_{r}-u_{ro}) & u_{ro}<u_{r}<u_{rm} \end{array}\right.$$
(3)$$Q(u_{b})=\left\{\begin{array}{cc} k_{be}u_{b} & 0<u_{b}<u_{bo}\\ k_{be}u_{bo}+k_{br}(u_{b}-u_{bo}) & u_{bo}<u_{b}<u_{bm} \end{array}\right.$$
where *k_rr_* and *k_re_* are the rigid and elastic parts of the bilinear function of the rail pad, *k_br_* and *k_be_* are the rigid and elastic parts of the bilinear function of the ballast, *u_r_* and *u_b_* are the rail pad and ballast deformation due to loads *Q*(*u_r_*) and *Q*(*u_b_*), respectively, *u_ro_* and *u_bo_* are the rail pad and ballast deformation when it is passing from the elastic to the rigid part, and *u_rm_* and *u_bm_* are the rail pad and ballast deformation for the maximum value of the load. Applying the least mean squares method, we obtain the parameters of the bilinear function. 

Figure 7 displays the bilinear functions of the rail pad and the ballast, calculated for a maximum force of 50 kN, and Table 1 presents the values of the parameters of the bilinear functions. 

Table 1 also contains the values of the stiffnesses of the two elastic layers calculated depending on a sleeper bay of 0.577 m:(4)$$\begin{array}{cccc} k_{1e}=\frac{k_{re}}{d}, & k_{1r}=\frac{k_{rr}}{d}, & k_{2e}=\frac{k_{be}}{d}, & k_{2r}=\frac{k_{br}}{d} \end{array},$$
where *k*_1*r*_ and *k*_1*e*_ are the stiffness of the rigid/elastic part of the bilinear function of the first elastic layer (rail pad), *k*_2*r*_ and *k*_2*e*_ are the stiffness of the rigid/elastic part of the bilinear function of the second elastic layer (ballast), and *d* is the sleeper bay. There are very large differences in the stiffness of the elastic part of the bilinear function associated with the load-deformation characteristic of the rail pad and small variations in the stiffness of its rigid part.

To evaluate the error introduced by the bilinear function and the range of applicability, the track equivalent characteristics must be calculated. This represents the uniform distributed load, *q*, depending on the rail displacement (Figure 8).
(5)$$q(w)=\left\{\begin{array}{c} k_{1}w,\\ k_{2}(w-w_{12}),\\ k_{3}(w-w_{23}), \end{array}\right.\begin{array}{c} 0<w<w_{12}\\ w_{12}<w<w_{23}\\ w_{23}<w<w_{m} \end{array}$$
where *w* is the rail displacement and the equivalent stiffnesses are
(6)$$\begin{array}{ccc} k_{1}=\frac{k_{1e}k_{2e}}{k_{1e}+k_{2e}}, & k_{2}=\frac{k_{1e}k_{2r}}{k_{1e}+k_{2r}}, & k_{3}=\frac{k_{1r}k_{2r}}{k_{1r}+k_{2r}} \end{array},$$
where *k*_1,2,3_ are the equivalent stiffnesses corresponding to the elastic, moderate, and stiff portions of the uniform distributed load–rail displacement characteristic, and the rail displacement for the transition points, *w*_12_ and *w*_23_, are given by:$$\begin{array}{cc} w_{12}=\frac{k_{2e}u_{bo}}{k_{1}}=u_{bo}\left(1+\frac{k_{2e}}{k_{1e}}\right), & \end{array}$$
(7)$$w_{23}=w_{12}+\frac{k_{1e}u_{ro}-k_{2e}u_{bo}}{k_{2}}=u_{ro}\left(1+\frac{k_{e1}}{k_{r2}}\right)+u_{bo}\left(1-\frac{k_{2e}}{k_{2r}}\right)$$
with *w*_12_ < *w*_23_, because this considers that (see Figure 6 and Figure 7):(8)$$k_{e1}u_{ro}>k_{e2}u_{bo}.$$

By imposing the maximum admissible error, we obtain the maximum rail displacement. For instance, considering a maximum error of 5%, Figure 9 presents the distributed load versus the rail displacement for both the theoretical nonlinear characteristics and a bilinear approximation for the three characteristics of the rail pad. The main results are summarized in Table 2. The Rms errors are acceptable, i.e., smaller than 1.91%.

### 2.3. Equilibrium Position of the Rail

In this section, the rail equilibrium position under a static load is determined. There are two reasons for this: (a) to check the applicability domain of the bilinear approximation in the sense that the maximum displacement of the rail is not higher than the limit of the applicability determined in the previous section (Table 2); and (b) to identify the transition sections of the elastic foundation, i.e., to calculate the distances *l*_1_ and *l*_2_, respectively.

Figure 10 presents the calculation model derived from the one presented in Figure 4; the inertial layer of the sleepers is not displayed for reasons of simplicity, because its position is not relevant here. 

The equilibrium equation of the rail under a static load can be refined from the two equilibrium equations of the track by eliminating the inertial layer displacement as:(9)$$EI\frac{d^{4}w(x)}{dx^{4}}+k(x)w(x)=q(x),$$
where *w*(*x*) is the rail displacement, *k*(*x*) is the equivalent stiffness of the two elastic layers depending on coordinate *x* along the rail, and *q*(*x*) is the distributed load supported by the rail with *q*(*x*) = *Q*δ(*x*), where δ(.) is the Dirac delta function.

The equivalent stiffness has the following form:(10)$$k(x)=\left\{\begin{array}{c} k_{3},\\ k_{2},\\ k_{1}, \end{array}\;\right.\begin{array}{c} \left|x\right|\leq l_{1}\\ 0<\left|x\right|-l_{1}\leq l_{2}\\ l_{1}+l_{2}<\left|x\right|, \end{array}$$
where ±*l*_1_ and ±(*l*_1_ + *l*_2_) are the coordinates along the rail corresponding to the transition sections of the uniform distributed load–rail displacement characteristic.

The rail displacement at the transition section, i.e., *x* = ±*l*_1_ and *x* = ± (*l*_1_ + *l*_2_), takes the values:(11)$$\begin{array}{cc} w(\pm l_{1})=w_{12}, & w(\pm (l_{1}+l_{2}))=w_{23} \end{array}.$$

The boundary condition associated with Equation (9) is:(12)$$\underset{\left|x\right|\to \infty }{\lim}w(x)=0.$$

Considering only a half rail due to symmetry and applying the direct method, which implies that the rail to be segmented is in the concentrated load section and in the transition sections, the equations of equilibrium are obtained:(13)$$EI\frac{d^{4}w_{i}(x_{i})}{dx_{i}^{4}}+k_{i}w_{i}(x_{i})=0,\;i=1÷3,$$
where displacement *w_i_*(*x_i_*) is the displacement of the *i* rail segment in section *x_i_*, which is related to rail displacement *w*(*x*) as follows:(14)$$w(x)=\left\{\begin{array}{ll} w_{3}(x_{3})+w_{12}+\frac{k_{1e}u_{ro}-k_{2e}u_{bo}}{k_{2}}, & x=x_{3};\;x\in \left[0,l_{1}\right);\;\\ w_{2}(x_{2})+w_{12}\frac{k_{2}-k_{1}}{k_{2}}, & x=l_{1}+x_{2};\;x\in \left[l_{1},l_{2}\right);\;x_{2}\in \left[0,l_{2}\right)\;\\ w_{1}(x_{1}), & x=l_{1}+l_{2}+x_{1};\;x\in \left[l_{2},\infty \right);\;x_{1}\in \left[0,\infty \right)\; \end{array}\right.$$

The boundary conditions associated with Equation (13) are as follows:-at *x*_3_ = 0, the rail slope is null and the shear force equals –*Q*/2
(15)$$\begin{array}{cc} \frac{dw_{3}(0)}{dx_{3}}=0, & \frac{d^{3}w_{3}(0)}{dx_{3}^{3}}=\frac{Q}{2EI} \end{array},$$

-at *x*_3_ = *l*_1_ and *x*_2_ = 0, the rail displacement is *w*_23_
(16)$$\begin{array}{cc} w_{3}(l_{1})=w_{23}-w_{12}-\frac{k_{1e}u_{ro}-k_{2e}u_{bo}}{k_{2}}, & w_{2}(0)=w_{23}-w_{12}\frac{k_{2}-k_{1}}{k_{2}}; \end{array}$$

and the first three derivatives of the rail displacement are continuous functions:(17)$$\frac{d^{n}w_{3}(l_{1})}{dx_{3}^{n}}=\frac{d^{n}w_{2}(0)}{dx_{2}^{n}}\;\mathrm{for}n=1,\;2,\;3;$$

-at *x*_2_ = *l*_2_ and *x*_1_ = 0, the rail displacement is *w*_12_
(18)$$\begin{array}{cc} w_{2}(l_{2})=w_{12}\frac{k_{1}}{k_{2}}, & w_{1}(0)=w_{12}; \end{array}$$

and the continuity conditions must be met as above:(19)$$\frac{d^{n}w_{2}(l_{2})}{dx_{2}^{n}}=\frac{d^{n}w_{1}(0)}{dx_{1}^{n}}\;\mathrm{for}\;n=1,\;2,\;3.$$

The solutions to differential Equation (13) are:(20)$$\begin{array}{l} w_{3}(x_{3})=A_{1}e^{-\alpha_{3}x_{3}}\sin\alpha_{3}x_{3}+A_{2}e^{-\alpha_{3}x_{3}}\cos\alpha_{3}x_{3}+A_{3}e^{\alpha_{3}x_{3}}\sin\alpha_{3}x_{3}+A_{4}e^{\alpha_{3}x_{3}}\cos\alpha_{3}x_{3}\\ w_{2}(x_{2})=A_{5}e^{-\alpha_{2}x_{2}}\sin\alpha_{2}x_{2}+A_{6}e^{-\alpha_{2}x_{2}}\cos\alpha_{2}x_{2}+A_{7}e^{\alpha_{2}x_{2}}\sin\alpha_{2}x_{2}+A_{8}e^{\alpha_{2}x_{2}}\cos\alpha_{2}x_{2}\\ w_{1}(x_{1})=A_{9}e^{-\alpha_{1}x_{1}}\sin\alpha_{1}x_{1}+w_{12}e^{-\alpha_{1}x_{1}}\cos\alpha_{1}x_{1}, \end{array}$$
where *A_n_* with *n* =1÷9 are constants to be determined, and
(21)$$\alpha_{i}=\sqrt[{4}]{\frac{k_{i}}{4EI}}\;\mathrm{with}\;i=1,\;2,\;3.$$

We observe that the shape of *w*_1_(*x*_1_) accomplishes the boundary condition at *x*_1_ = 0 and for *x*_1_→∞.

Equation (20) is inserted in the boundary conditions (15)–(19), resulting 11 non-linear equations of unknowns *A_n_*, *n* = 1 ÷ 9 and *l*_1_ and *l*_2_. Applying the Newton-Raphson algorithm, we obtain the unknown values, including the distances *l*_1_ and *l*_2_. Finally, the load–rail displacement characteristic can be drawn, as seen in Figure 11. Under a static load of 100 kN, the rail displacement is 1.465 mm for the soft rail pad, 1.352 mm for the medium rail pad, and 1.239 mm when the rail pad is stiff; in all cases, the limit of applicability of the bilinear approximation is not exceeded. 

The results regarding distances *l*_1_ and *l*_2_ are listed in Table 3.

### 2.4. Equations of the Dynamic Behavior

Wheel–rail vibrations develop around the equilibrium position and may be described by the displacement of the wheel, Δ*u*(*t*), the displacement of the rail, Δ*w*(*x*,*t*), and the displacement of the sleepers, Δ*z*(*x*,*t*).

The equations of motion are as follows:-for the wheel
(22)$$M_{w}\Delta \ddot{u}(t)=-\Delta Q(t);$$

-for the track

(23)$$\begin{array}{l} EI\frac{\partial^{4}\Delta w(x,t)}{\partial x^{4}}+m\frac{\partial^{2}\Delta w(x,t)}{\partial t^{2}}+k_{I}(x)\left[\Delta w(x,t)-\Delta z(x,t)\right]=\Delta Q(t)\delta (x)\\ m_{s}\frac{\partial^{2}\Delta z(x,t)}{\partial t^{2}}+\left[k_{I}(x)+k_{\mathrm{II}}(x)\right]\Delta z(x,t)-k_{I}(x)\Delta w(x,t)=0, \end{array}$$
where *k*_I,II_(*x*) is the stiffness of the *i* elastic layer (*i* = 1,2)
(24)$$k_{I}(x)=\left\{\begin{array}{ll} k_{1r}, & \left|x\right|\leq l_{1};\;\\ k_{1e}, & l_{1}<\left|x\right|\leq l_{2};\;\\ k_{1e}, & l_{1}+l_{2}<\left|x\right|;\; \end{array}\right.\;k_{\mathrm{II}}(x)=\left\{\begin{array}{ll} k_{2r}, & \left|x\right|\leq l_{1};\;\\ k_{2r}, & l_{1}<\left|x\right|\leq l_{2};\;\\ k_{2e}, & l_{1}+l_{2}<\left|x\right|,\; \end{array}\right.$$
and Δ*Q*(*t*) is the dynamic component of the contact force.

The following contact equation should be considered:(25)$$\Delta Q(t)=k_{H}\left[\Delta u(t)-\Delta w(0,t)-r(t)\right],$$
where *k_H_* is the contact stiffness, and *r*(*t*) is the irregularity of the rolling surfaces.

Considering the harmonic steady-state behavior, the equations of motion can be written as:(26)$$\omega^{2}M_{w}\Delta \bar{u}=\Delta \bar{Q}$$
(27)$$\begin{array}{l} EI\frac{d^{4}\Delta \bar{w}(x)}{dx^{4}}-\omega^{2}m\Delta \bar{w}(x)+{\bar{k}}_{I}(x)\left[\Delta \bar{w}(x)-\Delta \bar{z}(x)\right]=\Delta \bar{Q}\delta (x)\\ -\omega^{2}m_{s}\Delta \bar{z}(x)+\left[{\bar{k}}_{I}(x)+{\bar{k}}_{\mathrm{II}}(x)\right]\Delta \bar{z}(x)-{\bar{k}}_{I}(x)\Delta \bar{w}(x)=0, \end{array}$$
and the contact equation as:(28)$$\Delta \bar{Q}={\bar{k}}_{H}\left[\Delta \bar{u}-\Delta \bar{w}(0)-\bar{r}\right],$$
where $\Delta \bar{u}$,$\Delta \bar{w}(x)$,$\Delta \bar{z}(x)$, $\Delta \bar{Q}$, and $\bar{r}$ are the complex amplitudes of the wheel, rail, sleeper, contact force, and roughness, and
(29)$$\begin{array}{cc} {\bar{k}}_{I,\mathrm{II}}(x)=k_{I,\mathrm{II}}(x)(1+i\eta_{I,\mathrm{II}}),\; & {\bar{k}}_{H}=k_{H}(1+i\eta_{H}), \end{array}$$
where η_I,II_ is the loss factor of the first and second elastic layer, and η*_H_* is the loss factor of the elastic contact, and *i*^2^ = −1.

Equation (27) is equivalent to
(30)$$\begin{array}{l} EI\frac{d^{4}\Delta \bar{w}(x)}{dx^{4}}+\bar{K}(x,\omega )\Delta \bar{w}(x)=\Delta \bar{Q}\delta (x)\\ \Delta \bar{z}(x)=\frac{{\bar{k}}_{I}(x)}{-\omega^{2}m_{s}+{\bar{k}}_{I}(x)+{\bar{k}}_{\mathrm{II}}(x)}\Delta \bar{w}(x), \end{array}$$
where
(31)$$\bar{K}(x,\omega )=\frac{\omega^{4}mm_{s}-\omega^{2}\left[m\left({\bar{k}}_{I}(x)+{\bar{k}}_{\mathrm{II}}(x)\right)+m_{s}{\bar{k}}_{I}(x)\right]+{\bar{k}}_{I}(x){\bar{k}}_{\mathrm{II}}(x)}{-\omega^{2}m_{s}+{\bar{k}}_{I}(x)+{\bar{k}}_{\mathrm{II}}(x)}.$$

In fact, the complex parameter $\bar{K}(x,\omega )$ has a similar shape to those in Equation (24):(32)$$\bar{K}(x,\omega )=\left\{\begin{array}{ll} {\bar{K}}_{3}(\omega ), & \left|x\right|\leq l_{1};\;\\ {\bar{K}}_{2}(\omega ), & l_{1}<\left|x\right|\leq l_{2};\;\\ {\bar{K}}_{1}(\omega ), & l_{1}+l_{2}<\left|x\right|.\; \end{array}\right.$$

Following a similar method as in the equilibrium case, the following differential equations are considered:(33)$$\frac{d^{4}\Delta \bar{w}(x)}{dx^{4}}+\frac{{\bar{K}}_{3}(\omega )}{EI}\Delta \bar{w}(x)=0\;\mathrm{for}\;\left|x\right|\leq l_{1};$$
(34)$$\frac{d^{4}\Delta \bar{w}(x)}{dx^{4}}+\frac{{\bar{K}}_{2}(\omega )}{EI}\Delta \bar{w}(x)=0\;\mathrm{for}\;l_{1}<\left|x\right|\leq l_{2};$$
(35)$$\frac{d^{4}\Delta \bar{w}(x)}{dx^{4}}+\frac{{\bar{K}}_{1}(\omega )}{EI}\Delta \bar{w}(x)=0\;\mathrm{for}\;l_{1}+l_{2}<\left|x\right|.$$

The characteristic equations associated to the above differential equations are:(36)$$\lambda^{4}+\frac{{\bar{K}}_{n}(\omega )}{EI}=0\;,\;n=1,\;2,\;3$$
that gives the eigenvalues
(37)$$\begin{array}{l} \lambda_{1,3}=\pm \left(a_{n}-ib_{n}\right)\\ \lambda_{2,4}=\pm \left(b_{n}+ia_{n}\right) \end{array}$$
where *a_n_* and *b_n_* are positive real numbers.

The complex amplitude of the rail is:(38)$$\Delta \bar{w}(x)=e^{-\left(a_{1}-ib_{1}\right)x}W_{1}+e^{-\left(b_{1}+ia_{1}\right)x}W_{2}+e^{\left(a_{1}-ib_{1}\right)x}W_{3}+e^{\left(b_{1}+ia_{1}\right)x}W_{4}\;\mathrm{for}\;\left|x\right|\leq l_{1};$$
(39)$$\Delta \bar{w}(x)=e^{-\left(a_{2}-ib_{2}\right)x}W_{5}+e^{-\left(b_{2}+ia_{2}\right)x}W_{6}+e^{\left(a_{2}-ib_{2}\right)x}W_{7}+e^{\left(b_{2}+ia_{2}\right)x}W_{8}\;\mathrm{for}\;l_{1}<\left|x\right|\leq l_{2};$$
(40)$$\Delta \bar{w}(x)=e^{-\left(a_{3}-ib_{3}\right)x}W_{9}+e^{-\left(b_{3}+ia_{3}\right)x}W_{10}\;\mathrm{for}\;l_{1}+l_{2}<\left|x\right|,$$
where *W_n_* with *n* = 1÷10 are constants to be determined. The last form meets the boundary condition for *x*→∞.

The following boundary conditions hold:-at *x* = 0, the rail slope is null due to the symmetry and the shear force amplitude equalling $-\Delta \bar{Q}/2$
(41)$$\begin{array}{cc} \frac{d\Delta \bar{w}(0)}{dx}=0, & \frac{d^{3}\Delta \bar{w}(0)}{dx^{3}}=\frac{\Delta \bar{Q}}{2EI} \end{array};$$

-at *x* = *l*_1_, the continuity conditions are imposed:

(42)$$\begin{array}{l} \Delta \bar{w}(l_{1}-0)=\Delta \bar{w}(l_{1}+0),\\ \frac{d^{n}\Delta \bar{w}(l_{1}-0)}{dx^{n}}=\frac{d^{n}\Delta \bar{w}(l_{1}+0)}{dx^{n}}\;\mathrm{for}\;n=1,2,3, \end{array}$$
where the left parts are calculated from Equation (38) and the right parts are calculated from Equation (39).

-at *x* = *l*_1_ + *l*_2_, the continuity conditions are imposed

(43)$$\begin{array}{l} \Delta \bar{w}(l_{1}+l_{2}-0)=\Delta \bar{w}(l_{1}+l_{2}+0),\\ \frac{d^{n}\Delta \bar{w}(l_{1}+l_{2}-0)}{dx^{n}}=\frac{d^{n}\Delta \bar{w}(l_{1}+l_{2}+0)}{dx^{n}}\;\mathrm{for}\;n=1,2,3, \end{array}$$
where the left parts are calculated from Equation (39) and the right parts from Equation (40).

Finally, the following set of linear algebraic equations are obtained from Equations (38)–(43)
(44)$$-\left(a_{1}-ib_{1}\right)W_{1}-\left(b_{1}+ia_{1}\right)W_{2}+\left(a_{1}-ib_{1}\right)W_{3}+\left(b_{1}+ia_{1}\right)W_{4}=0;$$
(45)$$-{\left(a_{1}-ib_{1}\right)}^{3}W_{1}-{\left(b_{1}+ia_{1}\right)}^{3}W_{2}+{\left(a_{1}-ib_{1}\right)}^{3}W_{3}+{\left(b_{1}+ia_{1}\right)}^{3}W_{4}=\frac{\Delta \bar{Q}}{2EI};$$
(46)$$\begin{array}{l} e^{-\left(a_{1}-ib_{1}\right)l_{1}}W_{1}+e^{-\left(b_{1}+ia_{1}\right)l_{1}}W_{2}+e^{\left(a_{1}-ib_{1}\right)l_{1}}W_{3}+e^{\left(b_{1}+ia_{1}\right)l_{1}}W_{4}-\\ e^{-\left(a_{2}-ib_{2}\right)l_{1}}W_{5}-e^{-\left(b_{2}+ia_{2}\right)l_{1}}W_{6}-e^{\left(a_{2}-ib_{2}\right)l_{1}}W_{7}-e^{\left(b_{2}+ia_{2}\right)l_{1}}W_{8}=0 \end{array}$$
(47)$$\begin{array}{l} -\left(a_{1}-ib_{1}\right)e^{-\left(a_{1}-ib_{1}\right)l_{1}}W_{1}-\left(b_{1}+ia_{1}\right)e^{-\left(b_{1}+ia_{1}\right)l_{1}}W_{2}+\left(a_{1}-ib_{1}\right)e^{\left(a_{1}-ib_{1}\right)l_{1}}W_{3}+\left(b_{1}+ia_{1}\right)e^{\left(b_{1}+ia_{1}\right)l_{1}}W_{4}+\\ \left(a_{2}-ib_{2}\right)e^{-\left(a_{2}-ib_{2}\right)l_{1}}W_{5}+\left(b_{2}+ia_{2}\right)e^{-\left(b_{2}+ia_{2}\right)l_{1}}W_{6}-\left(a_{2}-ib_{2}\right)e^{\left(a_{2}-ib_{2}\right)l_{1}}W_{7}-\left(b_{2}+ia_{2}\right)e^{\left(b_{2}+ia_{2}\right)l_{1}}W_{8}=0 \end{array}$$
(48)$$\begin{array}{l} {\left(a_{1}-ib_{1}\right)}^{2}e^{-\left(a_{1}-ib_{1}\right)l_{1}}W_{1}+{\left(b_{1}+ia_{1}\right)}^{2}e^{-\left(b_{1}+ia_{1}\right)l_{1}}W_{2}+{\left(a_{1}-ib_{1}\right)}^{2}e^{\left(a_{1}-ib_{1}\right)l_{1}}W_{3}+{\left(b_{1}+ia_{1}\right)}^{2}e^{\left(b_{1}+ia_{1}\right)l_{1}}W_{4}-\\ {\left(a_{2}-ib_{2}\right)}^{2}e^{-\left(a_{2}-ib_{2}\right)l_{1}}W_{5}-{\left(b_{2}+ia_{2}\right)}^{2}e^{-\left(b_{2}+ia_{2}\right)l_{1}}W_{6}-{\left(a_{2}-ib_{2}\right)}^{2}e^{\left(a_{2}-ib_{2}\right)l_{1}}W_{7}-{\left(b_{2}+ia_{2}\right)}^{2}e^{\left(b_{2}+ia_{2}\right)l_{1}}W_{8}=0 \end{array}$$
(49)$$\begin{array}{l} -{\left(a_{1}-ib_{1}\right)}^{3}e^{-\left(a_{1}-ib_{1}\right)l_{1}}W_{1}-{\left(b_{1}+ia_{1}\right)}^{3}e^{-\left(b_{1}+ia_{1}\right)l_{1}}W_{2}+{\left(a_{1}-ib_{1}\right)}^{3}e^{\left(a_{1}-ib_{1}\right)l_{1}}W_{3}+{\left(b_{1}+ia_{1}\right)}^{3}e^{\left(b_{1}+ia_{1}\right)l_{1}}W_{4}+\\ {\left(a_{2}-ib_{2}\right)}^{3}e^{-\left(a_{2}-ib_{2}\right)l_{1}}W_{5}+{\left(b_{2}+ia_{2}\right)}^{3}e^{-\left(b_{2}+ia_{2}\right)l_{1}}W_{6}-{\left(a_{2}-ib_{2}\right)}^{3}e^{\left(a_{2}-ib_{2}\right)l_{1}}W_{7}-{\left(b_{2}+ia_{2}\right)}^{3}e^{\left(b_{2}+ia_{2}\right)l_{1}}W_{8}=0 \end{array}$$
(50)$$e^{-\left(a_{2}-ib_{2}\right)l}W_{5}+e^{-\left(b_{2}+ia_{2}\right)l}W_{6}+e^{\left(a_{2}-ib_{2}\right)l}W_{7}+e^{\left(b_{2}+ia_{2}\right)l}W_{8}-e^{-\left(a_{3}-ib_{3}\right)l}W_{9}-e^{-\left(b_{3}+ia_{3}\right)l}W_{10}=0$$
(51)$$\begin{array}{l} -\left(a_{2}-ib_{2}\right)e^{-\left(a_{2}-ib_{2}\right)l}W_{5}-\left(b_{2}+ia_{2}\right)e^{-\left(b_{2}+ia_{2}\right)l}W_{6}+\left(a_{2}-ib_{2}\right)e^{\left(a_{2}-ib_{2}\right)l}W_{7}+\left(b_{2}+ia_{2}\right)e^{\left(b_{2}+ia_{2}\right)l}W_{8}+\\ \left(a_{3}-ib_{3}\right)e^{-\left(a_{3}-ib_{3}\right)l}W_{9}+\left(b_{3}+ia_{3}\right)e^{-\left(b_{3}+ia_{3}\right)l}W_{10}=0 \end{array}$$
(52)$$\begin{array}{l} {\left(a_{2}-ib_{2}\right)}^{2}e^{-\left(a_{2}-ib_{2}\right)l}W_{5}+{\left(b_{2}+ia_{2}\right)}^{2}e^{-\left(b_{2}+ia_{2}\right)l}W_{6}+{\left(a_{2}-ib_{2}\right)}^{2}e^{\left(a_{2}-ib_{2}\right)l}W_{7}+{\left(b_{2}+ia_{2}\right)}^{2}e^{\left(b_{2}+ia_{2}\right)l}W_{8}-\\ {\left(a_{3}-ib_{3}\right)}^{2}e^{-\left(a_{3}-ib_{3}\right)l}W_{9}-{\left(b_{3}+ia_{3}\right)}^{2}e^{-\left(b_{3}+ia_{3}\right)l}W_{10}=0 \end{array}$$
(53)$$\begin{array}{l} -{\left(a_{2}-ib_{2}\right)}^{3}e^{-\left(a_{2}-ib_{2}\right)l}W_{5}-{\left(b_{2}+ia_{2}\right)}^{3}e^{-\left(b_{2}+ia_{2}\right)l}W_{6}+{\left(a_{2}-ib_{2}\right)}^{3}e^{\left(a_{2}-ib_{2}\right)l}W_{7}+{\left(b_{2}+ia_{2}\right)}^{3}e^{\left(b_{2}+ia_{2}\right)l}W_{8}+\\ {\left(a_{3}-ib_{3}\right)}^{3}e^{-\left(a_{3}-ib_{3}\right)l}W_{9}+{\left(b_{3}+ia_{3}\right)}^{3}e^{-\left(b_{3}+ia_{3}\right)l}W_{10}=0, \end{array}$$
where *l* = *l*_1_ + *l*_2_.

The solution to the above system can be obtained numerically, depending on contact force complex amplitude $\Delta \bar{Q}$. Then, the amplitude of the rail is calculated by inserting *W*_1÷10_ into Equations (38)–(40).

Rail receptance is the ratio between the rail amplitude and contact force amplitude: (54)$${\bar{G}}_{r}(x,\omega )=\frac{\Delta \bar{w}(x)}{\Delta \bar{Q}}.$$

Taking $\Delta \bar{Q}=1$, the rail receptance may be determined using Equations (44)–(54).

Similarly, the wheel receptance is the ratio between the wheel amplitude and the force amplitude
(55)$${\bar{G}}_{w}(\omega )=\frac{\Delta \bar{u}}{\Delta \bar{F}}=-\frac{1}{\omega^{2}M_{w}},$$
where the force and the wheel displacement have the same orientation. 

Considering the above, the wheel/rail vibrations are described by the following equations:(56)$$\Delta \bar{u}=-{\bar{G}}_{w}(\omega )\Delta \bar{Q};$$
(57)$$\Delta \bar{w}(0)={\bar{G}}_{r}(0,\omega )\Delta \bar{Q};$$
(58)$$\Delta \bar{u}-\Delta \bar{w}(0)-\bar{r}={\bar{G}}_{H}(\omega )\Delta \bar{Q},$$
where ${\bar{G}}_{H}(\omega )=1/{\bar{k}}_{H}$ is the wheel/rail contact receptance. 

Generally, the receptance is a complex number depending on the angular frequency, and because of this, all receptances have been notated corresponding to these features. However, the wheel receptance is a real number and the wheel/rail contact receptance does not depend on angular frequency according to the model.

The solution to Equations (56)*–*(58) is:(59)$$\begin{array}{ccc} \Delta \bar{Q}={\bar{H}}_{\Delta \bar{Q}}\bar{r}, & \Delta \bar{u}={\bar{H}}_{\Delta \bar{u}}\bar{r}, & \Delta \bar{w}(0)={\bar{H}}_{\Delta \bar{w}}\bar{r}, \end{array}$$
where
(60)$$\begin{array}{ccc} {\bar{H}}_{\Delta \bar{Q}}=-\frac{1}{{\bar{G}}_{w}(\omega )+{\bar{G}}_{r}(0,\omega )+{\bar{G}}_{H}(\omega )}, & {\bar{H}}_{\Delta \bar{w}}=-\frac{{\bar{G}}_{r}(0,\omega )}{{\bar{G}}_{w}(\omega )+{\bar{G}}_{r}(0,\omega )+{\bar{G}}_{H}(\omega )}, & {\bar{H}}_{\Delta \bar{u}}=\frac{{\bar{G}}_{w}(\omega )}{{\bar{G}}_{w}(\omega )+{\bar{G}}_{r}(0,\omega )+{\bar{G}}_{H}(\omega )} \end{array}$$
are the frequency response functions.

When it is assumed that the roughness of the rolling surfaces is a stationary and ergodic process described by the power spectral density *S_r_*(Ω), where Ω is the wavenumber, then the power spectral density depending on the angular frequency ω = *V*Ω is:(61)$$\Gamma_{r}(\omega )=\frac{S_{r}(\Omega )}{V}=\frac{S_{r}(\omega /V)}{V}.$$

The power spectral density of the output quantity *p* (Δ*u*, Δ*w* and Δ*Q*) that describes the vibration behavior is:(62)$$\Gamma_{p}(\omega )={\left|{\bar{H}}_{p}(\omega )\right|}^{2}\Gamma_{r}(\omega ),$$
where ${\bar{H}}_{p}(\omega )$ is the frequency response function associated with the quantity *p*, and the rms value calculated within an angular frequency interval is: (63)$$p_{rms}(\omega_{c})=\int_{\omega_{i}}^{\omega_{s}}\Gamma_{p}(\omega )d\omega ,$$
where ω*_i_* and ω*_s_* are the interval limits and ω*_c_* the central angular frequency.

## 3. Numerical Application

In this section, the wheel–rail model with a nonhomogeneous foundation is used to identify the impact of the variability of the rail pad elastic characteristics upon the wheel–rail vibrations, employing the MATLAB environment. 

Table 3 includes the values of the wheel–rail model parameters. The track is fitted with UIC 60 rail type, concrete sleepers of 268 kg, and the sleeper spacing is 577 mm. Stiffness values of the two elastic layers are the result of approximating the characteristics of the rail pad and the ballast with the help of the bilinear function, as shown in Section 2.1. The wheel mass and static load correspond to an electric locomotive.

Figure 12 shows the receptance modulus of the rail at the active point, i.e., where the harmonic force acts, as calculated for the homogenous two-layer model (Figure 13), where the stiffness of the elastic layers corresponds to the medium characteristic case (rigid part—*k*_1*r*_ = 443.464 MN/m^2^, *k*_2*r*_ = 96.681 MN/m^2^). This stiffness configuration is typically for the zone around the wheel–rail contact point, as shown in Figure 4. Considering the undamped case, the rail receptance exhibits two resonance frequencies, a low resonance at 91 Hz and a high one at 482 Hz. Between them, an antiresonance frequency at 240 Hz occurs due to the dynamic absorber effect of the sleepers. Conventionally, there are three ranges of frequency: the low frequency range, lower than the low resonance frequency, the medium frequency range between the two resonance frequencies, and the high frequency range located at frequencies higher than the high resonance frequency. The response of the rail is the effect of the dynamic balance of the forces acting on it, i.e., the elastic force, inertial force, and the excitation force. Thus, the rail receptance is almost constant at low frequency, because the elastic force is much greater than the inertial force, while at high frequency, the rail receptance decreases with frequency as the inertial force becomes dominant.

Figure 14 presents the rail receptance calculated for the nonhomogeneous two-layer track model for the three cases considered: soft, medium, and stiff characteristic. Additionally, the undamped case was considered. The rail receptance diagrams exhibit similar peaks, corresponding to the resonance frequencies indicated in the case of the homogeneous two-layer foundation, with small differences due to the differences between the stiffness values of the foundation around the active point in the three cases. However, these two peaks are flattened as an effect of the direct/reflected wave interference. In the range of low and high frequencies, the rail receptance shows similar characteristics to those highlighted in Figure 12. In the range of medium frequencies, the appearance of the rail receptance is different, being marked by abrupt variations due to the standing waves that are formed because of the overlapping of the direct bending waves with the reflected ones caused by the presence of the sections in which the foundation stiffness changes.

Figure 15 shows the impact of hysteretic damping on rail receptance. All abrupt variations of the rail receptance in the medium frequency range are removed due to the hysteretic damping. The two flattened peaks corresponding to the low and high resonance frequency and the depth characteristic of the anti-resonant behavior remain the same. Relative significant differences are maintained between the three cases considered in the field of low and average frequencies.

Figure 16 displays the frequency response function (F.R.F.) for the wheel displacement, rail displacement, and the contact force. At low frequency, the wheel takes the displacement imposed by the roughness of the rolling surfaces because the rail behaves as a rigid element and, due to this fact, the value of the F.R.F. module is around 1. The maximum value recorded at approx. 44 Hz is the effect of the resonance of the wheel on the track. At medium and high frequencies, the wheel displacement decreases because its receptance decreases as the frequency increases (see Equation (55)), and the rail takes the displacement imposed by the irregularity of the rolling surface, as can see in Figure 16b, where the F.R.F. modulus of the rail displacement is around 1. Regarding the response function of contact forces, this has a general increasing tendency depending on the frequency. It presents maxima relative to the wheel/track resonance frequency and to the anti-resonance frequency of the rail. At the two resonance frequencies of the rail, the contact force shows local minima. The impact of the rail pad elastic characteristic manifests itself mainly at medium frequencies and affects the wheel vibration and the magnitude of the contact force. Meanwhile, the rail vibration is less influenced by the variability of the elastic characteristic of the rail pad.

Figure 17 shows the spectrum of the rms value calculated for the wheel speed and rail speed at the active contact and for the contact force when the wheel velocity is 160 km/h. The spectrum is calculated in 1/3 octave intervals. The vibration speed was chosen as the parameter of interest because the acoustic power of the source depends on it, and the rolling noise is an important effect of wheel–rail vibrations. The data in Figure 17 are supplemented with those in Figure 18, where the ratio of the spectral components is presented. 

It is observed that the variability of the elastic characteristic of the rail pad influences the vibration of the wheel and the contact force in almost the entire considered frequency range. In most cases, the vibration of the wheel intensifies or decreases if the characteristic of the rail pad is stiff or soft. In general, the variations are smaller by a few percent, but there are also situations, especially with medium frequencies, where larger differences are observed. As for the rail, at very low and high frequencies, its vibrations are more intense with a soft rail pad. Otherwise, the rail vibration increases with a stiffer rail pad. In any case, the impact of the variability of the rail pad characteristic is less visible in the case of the rail.

## 4. Conclusions

Although at first glance, it seems to be a simple small piece of elastic material, the rail pad is an essential component of the track superstructure due to the role it plays in terms of vibration damping and reducing the shocks transmitted to the sleepers and ballast. 

The functionality of the rail pad depends on its most important property, namely, its load-deformation characteristics, which have a specific, nonlinear shape, and which present significant variability in mass production.

In this paper, the impact of the variability of the load-deformation characteristics of the rail support on the wheel–rail vibration behavior is studied. To this end, the nonlinear load-deformation characteristics have been approximated with the help of the bilinear function and implemented in a track model with a nonhomogeneous two-layer elastic foundation. 

The advantage of this method is that the model of the track with a nonhomogeneous foundation with continuous variation due the nonlinear characteristic of the rail pad and ballast is transformed into a model with nonhomogeneous foundation with two-step variation. In other words, the track model has three distinct portions with a homogeneous foundation, which simplifies the mathematical solution of the equations that describe the equilibrium state and the dynamic behavior. 

The load-deformation characteristic of the rail pad shows great variability in terms of the stiffness of the elastic part, while the stiffness of the rigid part is much more stable.

The rail pad works under the static load on the rigid side of the bilinear load-deformation characteristic, which has small deviations; this explains the relatively low impact of the variability of the rail pad characteristic on the wheel–rail vibrations. 

The results show that the impact of the variability of the load-deformation characteristic of the rail pad is most pronounced on the wheel–rail contact force and on the wheel vibrations, especially under the resonance frequencies of the wheel on the track and in the mid-frequency range. In contrast, rail vibrations have less influence. 

Future research will focus on expanding the frequency domain of the investigation by adapting the bilinear characteristic of the rail pad to the track model with discrete supports.

## Figures and Tables

**Figure 1 materials-16-01531-f001:**
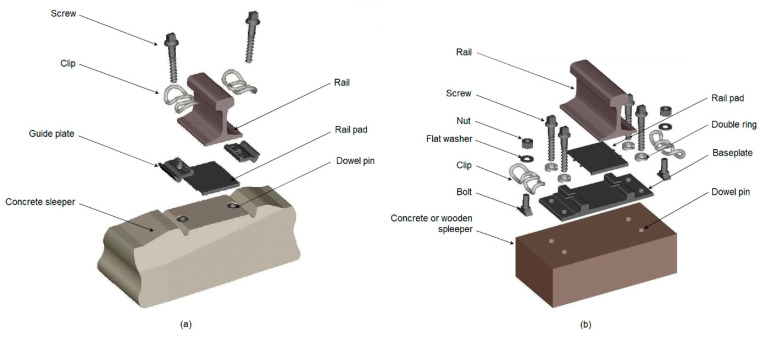
Fastening systems: (**a**) direct fastening; (**b**) indirect fastening (based on [1]).

**Figure 2 materials-16-01531-f002:**
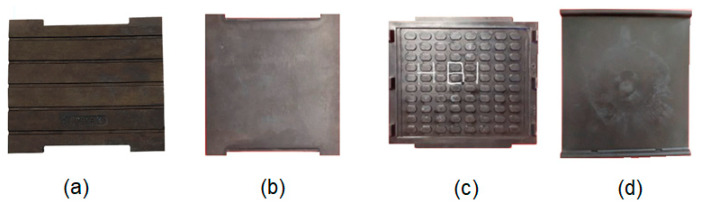
Rail pads: (**a**) rubber rail pad [4]; (**b**) EPDM rail pad; (**c**) TPE rail pad; (**d**) EVA rail pad; (**b**–**d**) based on [3].

**Figure 3 materials-16-01531-f003:**
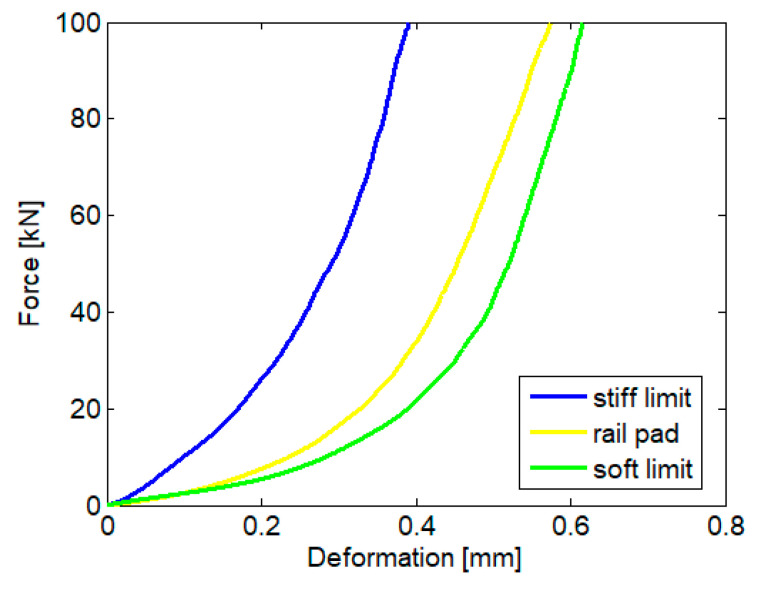
Load-deformation characteristics of a rail pad and the limit characteristics [39].

**Figure 4 materials-16-01531-f004:**
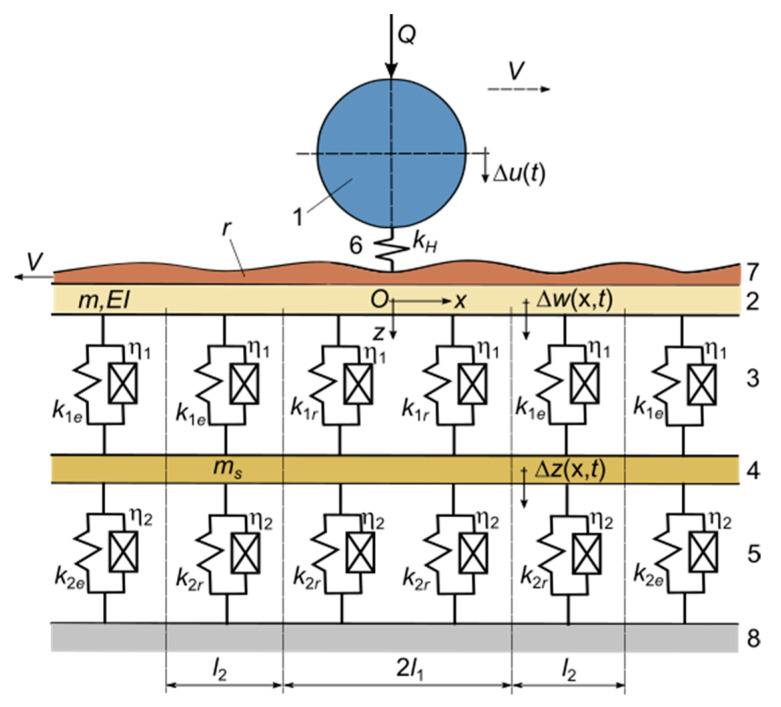
The mechanical model for the study of wheel–rail vibrations: (1) the wheel; (2) rail; (3) rail pads; (4) sleepers; (5) ballast prism; (6) elastic wheel–rail contact; (7) roughness; (8) rigid base.

**Figure 5 materials-16-01531-f005:**
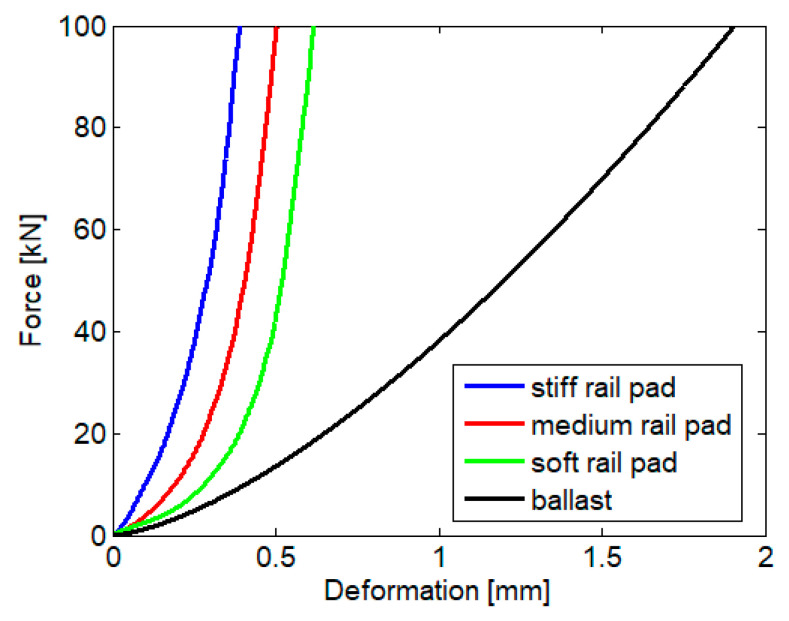
Nonlinear characteristic of the rail pad (based on [40]) and ballast (based on [5]).

**Figure 6 materials-16-01531-f006:**
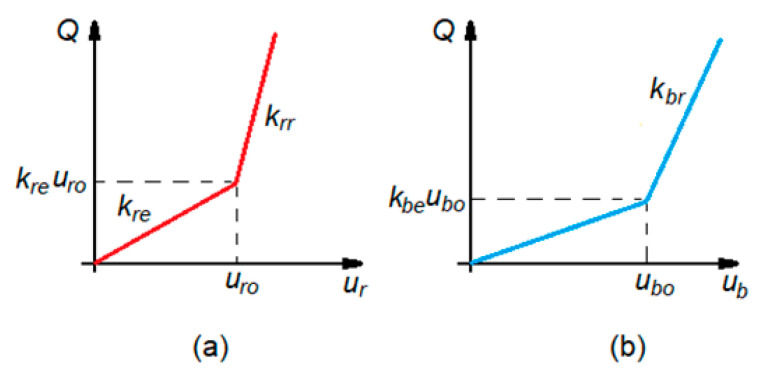
Bilinear function: (**a**) for rail pad; (**b**) for ballast.

**Figure 7 materials-16-01531-f007:**
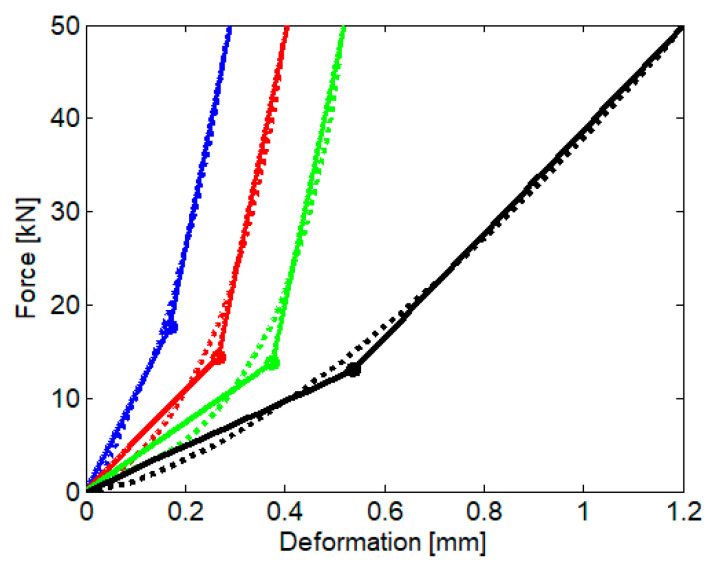
Bilinear approximation of the load-deformation characteristic: color code as in Figure 5; dotted lines—nonlinear function; continuous lines—bilinear approximation.

**Figure 8 materials-16-01531-f008:**
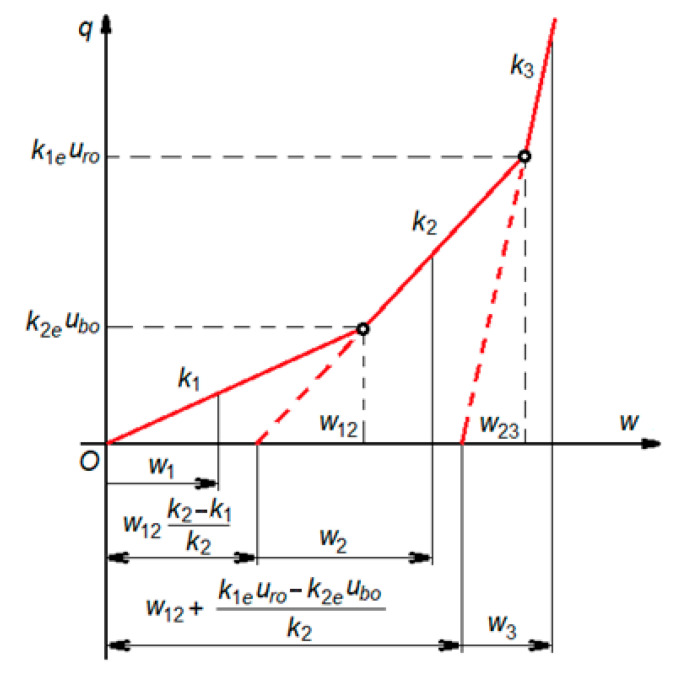
Equivalent characteristics.

**Figure 9 materials-16-01531-f009:**
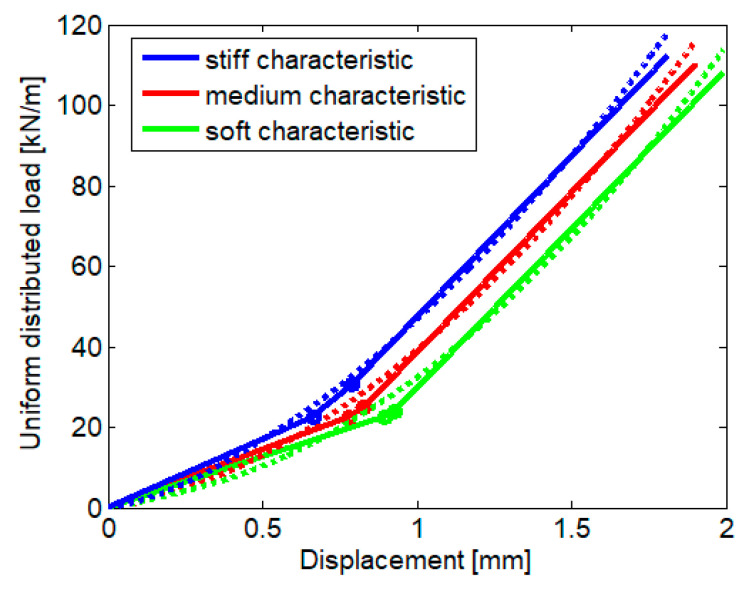
Equivalent characteristics of the track.

**Figure 10 materials-16-01531-f010:**
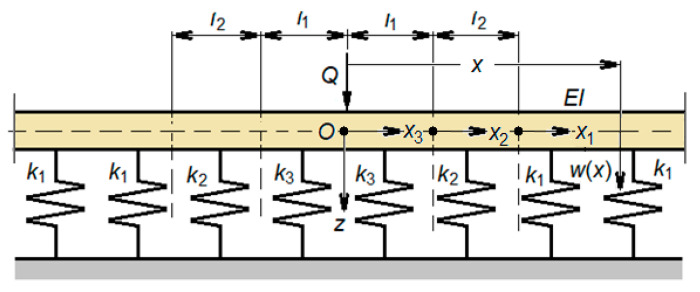
Explanation of the equilibrium position.

**Figure 11 materials-16-01531-f011:**
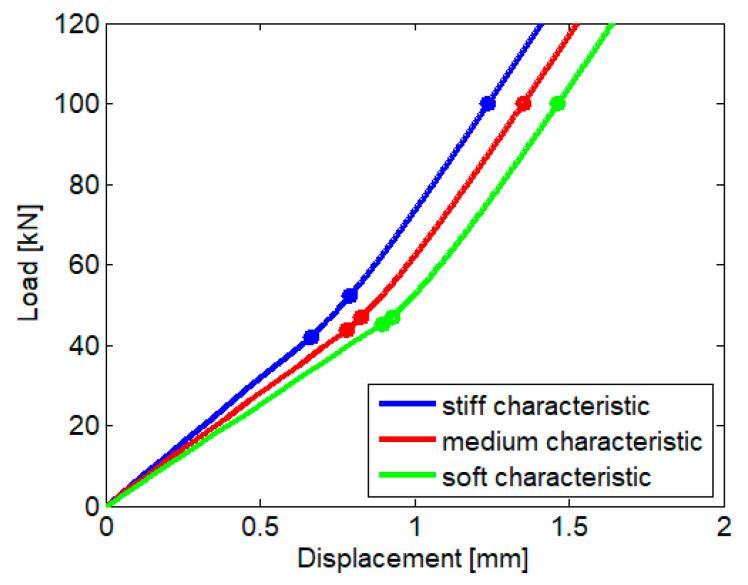
Load–rail displacement characteristic.

**Figure 12 materials-16-01531-f012:**
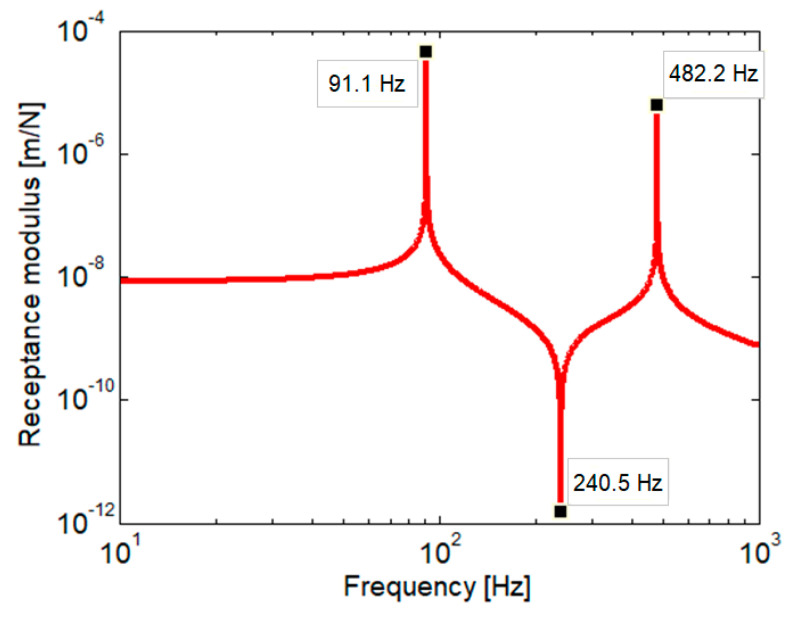
Rail receptance (homogeneous two-layer foundation).

**Figure 13 materials-16-01531-f013:**
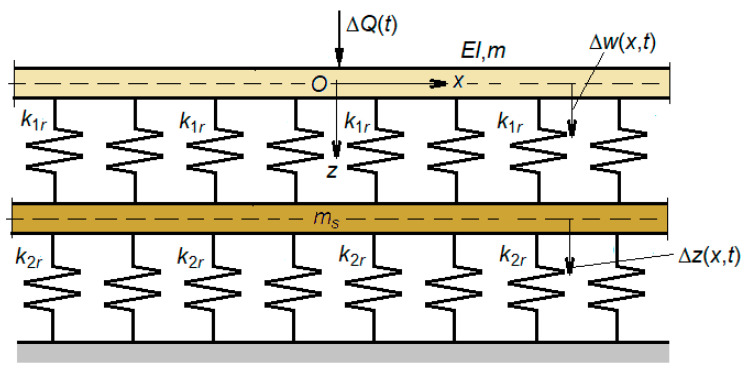
Homogeneous two-layer model of the track.

**Figure 14 materials-16-01531-f014:**
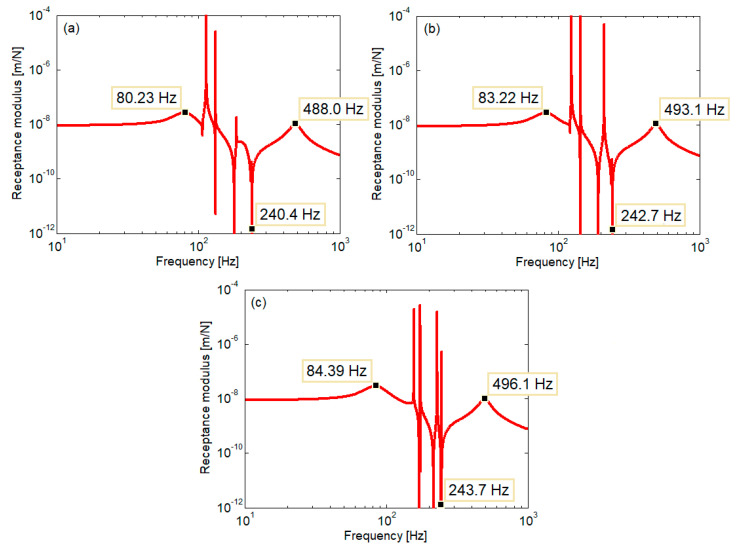
Rail receptance (nonhomogeneous two-layer foundation): (**a**) soft characteristic; (**b**) medium characteristic; (**c**) stiff characteristic.

**Figure 15 materials-16-01531-f015:**
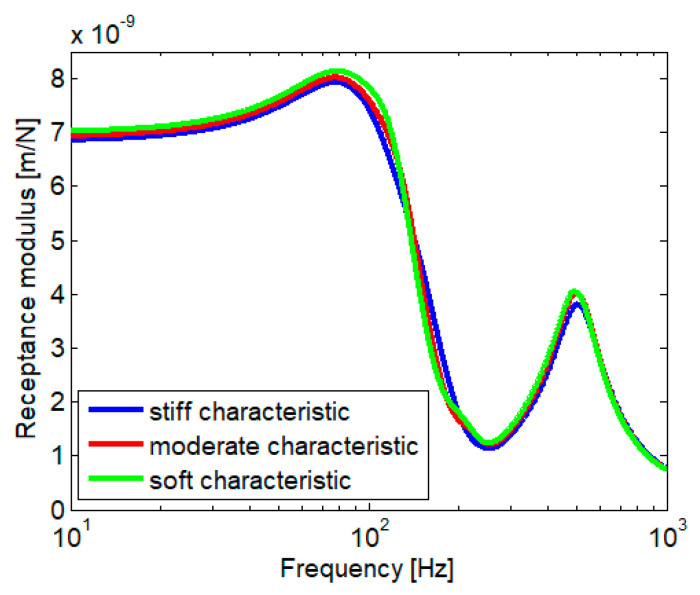
Rail receptance for the damped case.

**Figure 16 materials-16-01531-f016:**
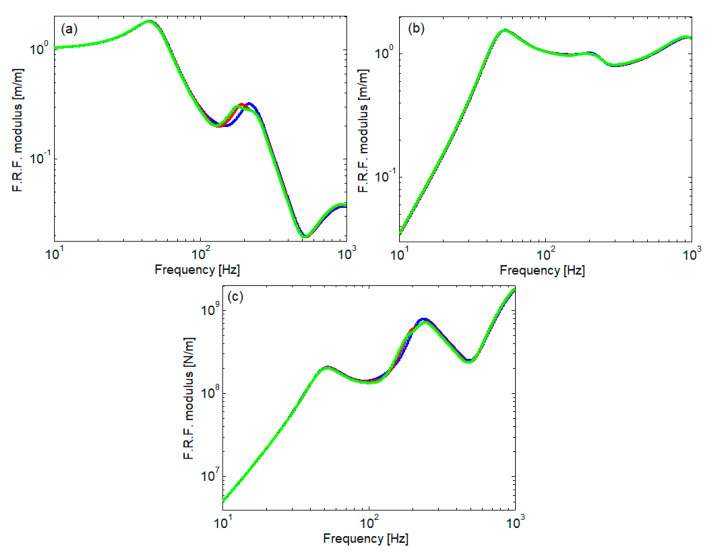
Frequency Response Function modulus: (**a**) wheel displacement; (**b**) rail displacement; (**c**) contact force; color code as in Figure 15.

**Figure 17 materials-16-01531-f017:**
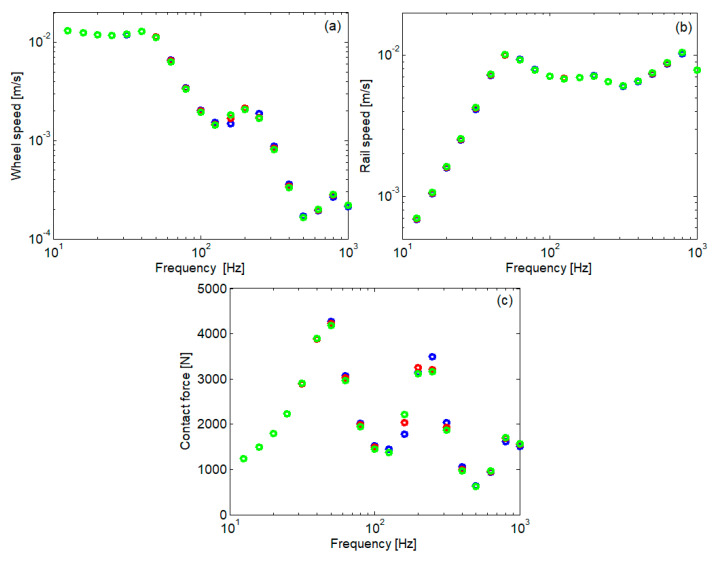
Spectrum of rms value: (**a**) wheel speed; (**b**) rail speed; (**c**) contact force; °, stiff characteristic, °, medium characteristic; °, soft characteristic; color code as in Figure 15.

**Figure 18 materials-16-01531-f018:**
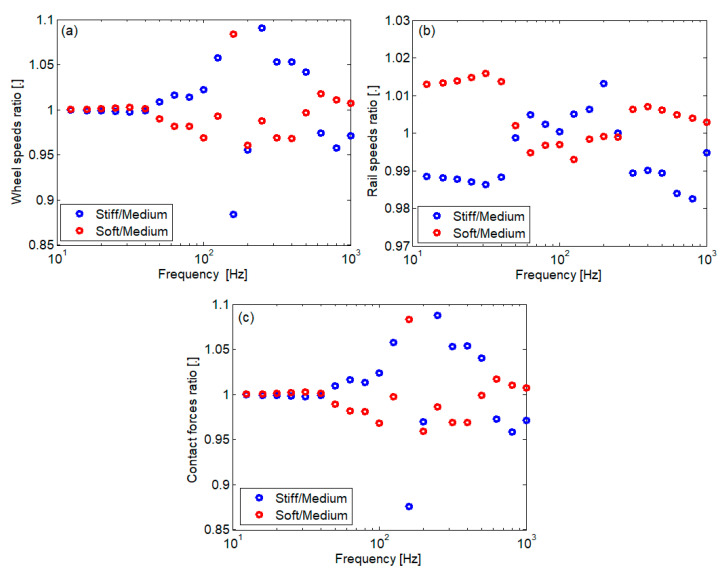
Spectral component ratios: (**a**) wheel speed; (**b**) rail speed; (**c**) contact force.

**Table 1 materials-16-01531-t001:** Parameters of the bilinear approximation.

Rail Pad Characteristic	*k_re_* [kN/mm]	*k_rr_* [kN/mm]	*k_be_* [kN/mm]	*k_br_* [kN/mm]	*k*_1*e*_ [MN/m^2^]	*k*_1*r*_ [MN/m^2^]	*k*_2*e*_ [MN/m^2^]	*k*_2*r*_ [MN/m^2^]
Soft rail pad	36.614	250.136	24.136	55.785	63.456	433.511	41.830	96.681
Medium rail pad	53.845	255.879			93.319	443.464		
Stiff rail pad	104.318	265.329			180.794	459.842		

**Table 2 materials-16-01531-t002:** Range of applicability and the errors of the bilinear approximation.

Rail Pad Characteristic	Maximum Error [%]	Rms Error [%]	Maximum Displacement [mm]
Soft rail pad	4.996	1.910	1.994
Medium rail pad	5.005	1.805	1.903
Stiff rail pad	4.997	1.626	1.815

**Table 3 materials-16-01531-t003:** Wheel–rail model parameters.

Parameter	Notation	Value	
Rail linear mass	*m*	60 kg/m	
Young’s modulus	*E*	210 GPa	
Area moment of inertia	*I*	30.55 × 10^−6^ m^4^	
Bending stiffness	*EI*	6.416 MNm^2^	
Sleeper linear mass (half)	*m_s_*	232 kg/m	
Rail pad stiffness (first layer)	*k* _1*e*_	Soft	63.456 MN/m^2^
		Medium	93.319 MN/m^2^
		Stiff	180.794 MN/m^2^
	*k* _1*r*_	Soft	433.511 MN/m^2^
		Medium	443.464 MN/m^2^
		Stiff	459.842 MN/m^2^
Rail pad loss factor	η_1_	0.25	
Hertz constant (ballast charact.)	*C_H_*	1.2·10^9^ Nm^2/3^	
Ballast stiffness (second layer)	*k* _2*e*_	41.830 MN/m^2^	
	*k* _2*r*_	96.681 MN/m^2^	
Ballast loss factor	η_2_	1	
Distance	*l* _1_	Soft 779 mm	
		Medium 783 mm	
		Stiff 717	
	*l* _2_	Soft 34 mm	
		Medium 53 mm	
		Stiff 148 mm	
Wheel mass	*M_w_*	1200 kg	
Static load	*Q*	100 kN	
Contact stiffness	*k_H_*	1.524 GN/m	
Contact loss factor	η*_H_*	0	

## Data Availability

Not applicable.

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
