# Peer review of "Parametric Study of the Influence of Nonlinear Elastic Characteristics of Rail Pads on Wheel–Rail Vibrations"

_materials, 2023, doi:10.3390/ma16041531_

Round 1

Reviewer 1 Report

The following issues should be addressed before the publication of the article.

1. All parameters in the equations should be explained. Besides, different symbols should be used to distinguish the different parameters.

2.  The advantage of the bilinear function should be provided.

3. Texts in Fig. 12 and 14 is too small. 

4.  Number of digits following the decimal point should be same.

5. A brief description for values of parameters in Table 3 should be added. 

Author Response

Reply to #1 Reviewer

First, we would like to thank you for taking the time to review the paper and for your comments. The original comments and our responses are listed below. We hope that we have adequately addressed all the comments and that our answers are satisfactory.

  1. All parameters in the equations should be explained. Besides, different symbols should be used to distinguish the different parameters.

Reply: We revised, including Fig. 10 for clarity.

  1. Number of digits following the decimal point should be same.

Reply: We revised.

  1. The advantage of the bilinear function should be provided.

Reply: The following text has been added in Conclusions: The advantage of this approach is that the model of the track with nonhomogeneous foundation with continuous variation due to the nonlinear elastic characteristic of the rail pad and ballast is transformed into a model with nonhomogeneous foundation with two-step variation. In other words, the track model has three distinct portions with homogeneous foundation, which simplifies the mathematical solution of the equations that describe the equilibrium state and the dynamic behaviour.

  1. Texts in Fig. 12 and 14 is too small.

Reply: We revised Figs. 12 and 14.

  1. A brief description for values of parameters in Table 3 should be added. 

Reply: Lines 428-433 (revised version) contain a brief description for values of parameters, which has been completed (red text): “Table 3 includes the values of the wheel-rail model parameters. The track is fitted with UIC 60 rail type, concrete sleepers of 268 kg, and the sleeper spacing is 577 mm. Stiffness values of the two elastic layers are the result of approximating the characteristics of the rail pad and the ballast with the help of the bilinear function, as shown in section 2.1. Wheel mass and static load correspond to an electric locomotive.”

Reviewer 2 Report

- The manuscript would benefit from proof reading proof reading: there is a number of places where a word is split with hyphen (see for instance line 69 on Page2 meth-od);

- It is not clear how the authors moved from the dynamic system shown in Figure 4 to the system shown in Figure 10; 

- The authors should consider splitting the Section 2 into several subsections - for instance Force displacement approximation, Static solution and dynamic solution;  

- The authors did not state which software was used for the model implementation and results generation;  

- It would be good to separate static and dynamic results and provide the discussion separately, including the comments in the Conclusion.  

Author Response

Reply to #2 Reviewer

First, we would like to thank you for taking the time to review the paper and for your comments. The original comments and our responses are listed below. We hope that we have adequately addressed all the comments and that our answers are satisfactory.

  1. The manuscript would benefit from proof reading proof reading: there is a number of places where a word is split with hyphen (see for instance line 69 on Page2 meth-od).

Reply: We revised.

  1. It is not clear how the authors moved from the dynamic system shown in Figure 4 to the system shown in Figure 10.

Reply: Fig. 4 shows the model for the study of wheel-rail vibrations of the wheel/rail. This includes the wheel which is loaded with the static load Q which is the weight of the suspended mass of vehicle on wheel and the weight of the wheel itself and the track. Fig. 10 presents the equilibrium position of the wheel/rail system which it is equivalent to the rail under the static load (applied by the wheel). Rail position under static load depends only on the equivalent stiffness of the two elastic layers which work in series. In this matter, the position of the inertial layer of the sleepers is not important and because of that this layer has not displayed in Fig. 10. From mathematical viewpoint, there are two equations of equilibrium, one for the rail and other one for the inertial layer position. Eliminating the displacement of the inertial layer, the rail equation results depending only on the equivalent stiffness of the two elastic layers. We have added the necessary explanations in the text.

  1. The authors should consider splitting the Section 2 into several subsections - for instance Force displacement approximation, Static solution and dynamic solution;.

Reply: We revised the Section 2 which has been split into: 2.1. Hypotheses and structure of the mechanical model; 2.2. Bilinear approximation of the elastic characteristics of the rail pad and ballast; 2.3. Equilibrium position of the rail; 2.4. Equations of the dynamic behaviour.

On the other hand, the following comment has been added in Conclusions: The advantage of this approach is that the model of the track with nonhomogeneous foundation with continuous variation due the nonlinear elastic characteristic of the rail pad and ballast is transformed into a model with nonhomogeneous foundation with two-step variation. In other words, the track model has three distinct portions with homogeneous foundation, which simplifies the mathematical solution of the equations that describe the equilibrium state and the dynamic behaviour.

  1. The authors did not state which software was used for the model implementation and results generation.

Reply: We added: …. employing the MATLAB environment (line 428).

  1. It would be good to separate static and dynamic results and provide the discussion separately, including the comments in the Conclusion. brief description for values of parameters in Table 3 should be added.

Reply: Restructuring the Section 2, the requirement has been accomplished as following: Section 2.3 deals with the equilibrium position of the rail under the static load applied by the wheel to check the applicability domain of the bilinear approximation in the sense that the maximum displacement of the rail is not higher than the limit of applicability determined in the previous section (2.2) and to identify the transition sections of the elastic foundation, respectively the calculation of the distances l1 and l2. All dynamic results are presented in Section 3. Conclusion has been rearranged to meet the reviewer’s requirement.

Reviewer 3 Report

Referee Report

on paper “ Parametric study of the influence of non-linear elastic characteristics of rail pads on wheel-rail vibration “ (materials-2181450) by author Traian Mazilu, Mădălina Dumitriu and IonuÈ›-Radu Răcănel submitted to Materials

This is interesting theoretical paper. It reports the problem of the influence of the variability of the nonlinear elastic characteristic of the rail pad resulting from the manufacturing process, on the wheel-rail vibrations. All the elastic characteristics have been approximated with the help of the bilinear function and are implemented in the track model consisting of a beam Euler-Bernoulli on a continuous foundation with two elastic layers and inertial insertion, resulting in a model with inhomogeneous foundation. The parameters of the inhomogeneous foundation have been established from the equilibrium condition under the static load. The presented results are reliable without any doubts. However, I have some comments and additions. I would like to note a few points to improve the paper before it can be published:

1.    All the motivation should be deleted from the Abstract.

2.    The authors should mention in 1. Introduction some information about composite materials are perspective for practical applications:

(1). M.A. Almessiere, Y. Slimani, N.A. Algarou, M.G. Vakhitov, D.S. Klygach, A. Baykal, T.I. Zubar, S.V. Trukhanov, A.V. Trukhanov, H. Attia, M. Sertkol, I.A. Auwal, Tuning the structure, magnetic and high frequency properties of Sc-doped Sr0.5Ba0.5ScxFe12-xO19/NiFe2O4 hard/soft nanocomposites, Adv. Electr. Mater. 8 (2022) 2101124. https://doi.org/10.1002/aelm.202101124.

3.    The authors should mention in 1. Introduction such experimental methods of non-destructive testing and determination of microstresses in materials as X-ray or/and neutron diffraction:

(2). S.V. Trukhanov, A.V. Trukhanov, V.G. Kostishyn, L.V. Panina, V.A. Turchenko, I.S. Kazakevich, An.V. Trukhanov, E.L. Trukhanova, V.O. Natarov, A.M. Balagurov, Thermal evolution of exchange interactions in lightly doped barium hexaferrites, J. Magn. Magn. Mater. 426 (2017) 554-562. http://dx.doi.org/10.1016/j.jmmm.2016.10.151.

4.    The authors should mention in 1. Introduction some experimental methods for assessing surface tension, friction and wear in materials:

(3). M.A. Darwish, T.I. Zubar, O.D. Kanafyev, D. Zhou, E.L. Trukhanova, S.V. Trukhanov, A.V. Trukhanov, A.M. Henaish, Combined effect of microstructure, surface energy, and adhesion force on the friction of PVA/ferrite spinel nanocomposites, Nanomaterals 12 (2022) 1998. https://doi.org/10.3390/nano12121998.

5.    The proposed 6 papers should be inserted in References.

The paper should be sent to me for the second analysis after the moderate revisions.

Author Response

Reply to #3 Reviewer

First, we would like to thank you for taking the time to review the paper and for your comments. The original comments and our responses are listed below.

  1. This is interesting theoretical paper. It reports the problem of the influence of the variability of the nonlinear elastic characteristic of the rail pad resulting from the manufacturing process, on the wheel-rail vibrations. All the elastic characteristics have been approximated with the help of the bilinear function and are implemented in the track model consisting of a beam Euler-Bernoulli on a continuous foundation with two elastic layers and inertial insertion, resulting in a model with inhomogeneous foundation. The parameters of the inhomogeneous foundation have been established from the equilibrium condition under the static load. The presented results are reliable without any doubts.

Reply: We would like to thank Reviewer III for his appreciation.

  1. The authors should mention in 1. Introduction some information about composite materials are perspective for practical applications:

(1). M.A. Almessiere, Y. Slimani, N.A. Algarou, M.G. Vakhitov, D.S. Klygach, A. Baykal, T.I. Zubar, S.V. Trukhanov, A.V. Trukhanov, H. Attia, M. Sertkol, I.A. Auwal, Tuning the structure, magnetic and high frequency properties of Sc-doped Sr0.5Ba0.5ScxFe12-xO19/NiFe2O4 hard/soft nanocomposites, Adv. Electr. Mater. 8 (2022) 2101124. https://doi.org/10.1002/aelm.202101124.

  1. The authors should mention in 1. Introduction such experimental methods of non-destructive testing and determination of microstresses in materials as X-ray or/and neutron diffraction:

(2). S.V. Trukhanov, A.V. Trukhanov, V.G. Kostishyn, L.V. Panina, V.A. Turchenko, I.S. Kazakevich, An.V. Trukhanov, E.L. Trukhanova, V.O. Natarov, A.M. Balagurov, Thermal evolution of exchange interactions in lightly doped barium hexaferrites, J. Magn. Magn. Mater. 426 (2017) 554-562. http://dx.doi.org/10.1016/j.jmmm.2016.10.151.

  1. The authors should mention in 1. Introduction some experimental methods for assessing surface tension, friction and wear in materials:

(3). M.A. Darwish, T.I. Zubar, O.D. Kanafyev, D. Zhou, E.L. Trukhanova, S.V. Trukhanov, A.V. Trukhanov, A.M. Henaish, Combined effect of microstructure, surface energy, and adhesion force on the friction of PVA/ferrite spinel nanocomposites, Nanomaterals 12 (2022) 1998. https://doi.org/10.3390/nano12121998.

  1. The proposed 6 papers should be inserted in References.

Reply: Our paper is about the influence of the variability of the nonlinear elastic characteristic of the rail pad resulting from the manufacturing process, on the wheel-rail vibrations is investigated. Rail pad is the elastic element between the rail and the sleeper that has the role of absorbing the mechanical stresses from the rail and reducing the vibrations and shocks generated by the wheel-rail interaction. Rail pad is made from elastic materials like rubber, ethylene propylene diene monomer (EPDM), thermoplastic polyester elastomer (TPE), ethylene-vinyl acetate (EVA) or high-density polyethylene (HDPE).

We observe that first paper is about the hard/soft nanocomposites, the second is about lightly doped barium hexaferrites and the third is about PVA/ferrite spinel nanocomposites. None of the three materials are used to manufacture the rail pad.

We draw attention to the fact that the MDPI evaluation policy of the papers includes the criterion "Are all the cited references relevant to the research?" that is mandatory, not derisive.

In conclusion, the papers recommended by the reviewer III have nothing in common with our paper and cannot be cited.
